# Association of Anxiety over Radiation Exposure and Acquisition of Knowledge Regarding Occupational Health Management in Operation Leader Candidates of Radioactivity Decontamination Workers in Fukushima, Japan: A Cross-Sectional Study

**DOI:** 10.3390/ijerph17010228

**Published:** 2019-12-28

**Authors:** Tomoo Hidaka, Takeyasu Kakamu, Shota Endo, Hideaki Kasuga, Yusuke Masuishi, Tomohiro Kumagai, Sei Sato, Takuma Sasaki, Tetsuhito Fukushima

**Affiliations:** Department of Hygiene and Preventive Medicine, Fukushima Medical University; Fukushima 960-1247, Japan

**Keywords:** occupational health management, occupational mental health management, knowledge management, risk perception, work safety, radioactivity decontamination work, Fukushima Nuclear Power Plant accident, disaster psychology, employee education

## Abstract

An operation leader (OL) of radioactivity decontamination workers is required by law to have accurate knowledge of occupational health management (OHM) such as working environment management, control of operations, and health management as a field supervisor in Japan. The purpose of the current study is to examine the association between anxiety over radiation exposure and the possession/acquisition of the OHM knowledge required for OLs. In this cross-sectional study, data were collected before and after training sessions held by Fukushima Prefecture Labor Standard Associations in Fukushima, Japan, in 2017. Eighty-seven males who completed the questionnaires were enrolled to this study. As a result, acquisition of knowledge of working environment management was significantly associated with an increase of anxiety over radiation exposure after the session comparing the situation before and after the session (knowledge possession; odds ratio = 4.489; 95% confidence interval = 1.216, 16.571). In conclusion, acquisition of accurate knowledge of how to manage working environment management was associated with anxiety over radiation exposure. Although acquisition of said knowledge may contribute to the reduction of physical health risks, it may increase mental health risks. Both mental health support for OLs with accurate knowledge and educational support for those without accurate knowledge are required.

## 1. Introduction

Eight years have passed since the Fukushima Daiichi Nuclear Power Plant accident in 2011. Following the accident, three evacuation areas were determined in Fukushima Prefecture by the Japanese government according to radiation dose rates in 2012: the “difficult-to-return zones” with 50 mSv/year or more; the “restricted residence zones” between 20 and 50 mSv/year; and the “evacuation order cancellation preparation zones” with 20 mSv/year or less [1,2]. Radioactivity decontamination work has started in the “difficult-to-return zones” and is now in progress, while said work was completed in the other two zone types by April 2017 [1].

To prevent physical and mental problems caused by radiation exposure, the occupational health management (OHM) of radioactivity decontamination workers in the “difficult-to-return zones” has grown in importance. An operation leader (OL) is a field supervisor and is required by law [3,4] to acquire accurate knowledge of the OHM. The OHM knowledge is specifically related to radioactivity decontamination work, such as the handling of radiation detectors, for the safety and health of decontamination workers.

For the association between radiation-related knowledge and anxiety, results from past studies have been inconsistent: said knowledge has been reported to be associated with both increase/presence of anxiety [5,6] and decrease/absence of anxiety [7,8,9,10,11]. Moreover, little is known of the association between such knowledge and anxiety in OLs.

The purpose of the current study is to examine the association between anxiety over radiation exposure and the possession/acquisition of the knowledge of OHM required for OLs. Considering past studies that reported that radiation-related knowledge possession and absence of anxiety were associated among professional workers [8], we hypothesized that possession of accurate knowledge regarding OHM is associated with absence of anxiety. We suspected that OLs’ accurate knowledge for OHM contributes to anxiety reduction, and that such anxiety reduction leads to the safety and security of radioactivity decontamination work; however, anxiety over radiation exposure is not necessarily entirely negative. Rather, such anxiety is possibly associated with effective avoidance of radiation exposure if the anxiety is backed by accurate knowledge.

## 2. Materials and Methods

### 2.1. Setting and Participants

Participants’ data were obtained from our previous study [12]. In the present cross-sectional study, a total of 103 candidates for OL positions were included. These candidates participated in training sessions on either April 7th, June 22nd, or August 18th 2017 held by Fukushima Prefecture Labor Standard Associations. They were handed out two self-administered questionnaires at the beginning of training session, and were asked to return one after filling out, before and after the session. One hundred and two candidates returned the two questionnaires anonymously (96 males, four females, and two blanks; response rate 99.0%). Because nearly all the respondents were male, we included 87 males who completed the questionnaire in the final analysis as the subjects of the present study. The effective response rate was therefore 90.6% (87/96).

### 2.2. Measurements

The variables including age, anxiety over radiation exposure, and OHM knowledge required for OLs were measured using questionnaires. Of these variables, anxiety and knowledge were measured both before and after the session.

#### 2.2.1. Occupational Health Management Knowledge Required for OLs

The knowledge required for OLs was classified into three types: working environment management, control of operations, and health management. The subjects’ knowledge was assessed by the following questions, in the same procedure as that used in our previous study (questions shown in Appendix A) [12]: “Choose one incorrect explanation regarding the measurement of ambient dose rate at decontamination sites in the preliminary survey” for working environment management; “Choose one incorrect explanation regarding the appropriate use of protective equipment” for control of operations; and “Choose one incorrect method of prevention of heat illness” for health management. Health management knowledge was assessed by a question on heat illness, since heat illness has been proved to be a more common health problem than radiation exposure among radioactivity decontamination workers [13,14]. These questions were constructed using the law [4], guidelines [3], and a standard textbook used in the training sessions [15].

#### 2.2.2. Anxiety Over Radiation Exposure and Age

The degree of anxiety over radiation exposure was assessed using the following question from a past study: “How much anxiety do you have over radiation exposure?” The answers were measured on a four-point scale (1 = “very much,” 2 = “somewhat,” 3 = “a little bit,” and 4 = “none”) [12]. As the sociodemographic data, the participants were asked to provide their age, and they were then classified into five age groups: < 30, 30–39, 40–49, 50–59, and ≥ 60 years.

### 2.3. Statistical Analysis

Statistical analyses were performed in June 2019, using SPSS statistics version 25 (IBM Corp., Armonk, NY, USA). The participant’s characteristics were summarized using descriptive statistics. Prior to bivariate and multivariate analyses, anxiety over radiation exposure was dichotomized (“very much” and “somewhat” for “present”, and “a little bit” and “none” for “absent”), and two answers to each question of OHM obtained before and after the session were classified into the following four patterns according to the correctness of the answers: “correct to correct”, “correct to incorrect”, “incorrect to correct”, and “incorrect to incorrect”. Knowledge possession was defined as the pattern “correct to correct”, and knowledge acquisition was defined as the pattern “incorrect to correct”. The “correct to correct” pattern was set as the referent for the multivariate analysis.

In the bivariate analysis, associations of the answer patterns with anxiety over radiation exposure before and after the sessions were examined by chi-square test. Because no significant association was found between them, multivariate analysis was employed only for the association between the answer patterns and anxiety after the session.

We used a binary logistic regression analysis to examine the associations of explanatory variables such as answer patterns and age with objective variable such as anxiety over radiation exposure after the session using direct method. Then, odds ratio (OR) and 95% confidence interval (CI) for presence of anxiety over radiation exposure were calculated. Models were built in three steps: a crude model, in which the associations of anxiety over radiation exposure with age and knowledge of OHM were examined without adjustment; an age-adjusted model, in which the association of anxiety over radiation exposure and OHM knowledge was examined, inputting age consistently as an explanatory variable for adjustment; and lastly a multivariate-adjusted model which served as the final model in which all explanatory variables were input simultaneously and fully adjusted.

The variance inflation factor (VIF) was used to test multicollinearity for multivariate-adjusted model (Model 3). The VIFs for age, and knowledge of working environment management, control of operations, and health management were 1.048, 1.061, 1.025, and 1.008, respectively. None of the VIF values reached 10, and the overall mean VIF of the model was less than 6. Thus, there was no collinearity.

P values below 0.05 were regarded as statistically significant.

### 2.4. Ethics Approval

This study was approved by the Ethics Committees of Fukushima Medical University (Application No. 3035). For informed consent procedure, all study participants were explained the purpose of the current study and returned the questionnaires anonymously only when they provided the informed consent.

## 3. Results

As shown in Table 1, the mean age of the participants was 47.6 (standard deviation (SD): 12.5) and the largest age group was 50-59 years (35.6%). For answer patterns, the pattern “correct to correct” was consistently common for all three questions on working environment management (64.4%), control of operations (43.7%), and health management (43.7%). The pattern “incorrect to correct” was the second most common in working environment management and control of operations (19.5% and 27.6%, respectively).

Table 2 shows that no significant associations were found between anxiety over radiation exposure before the session and variables, although there was trend that the anxiety over radiation exposure was more prevalent in young individuals such as < 30 than other age groups. The P values of the answer patterns in knowledge of working environment management and of control of operations were 0.888 and 0.750, respectively.

As shown in Table 3, anxiety over radiation exposure after the training session was significantly associated with answer patterns in knowledge of working environment management (p = 0.040). No significant associations were found between anxiety over radiation exposure after training session and other variables such as answer patterns in knowledge of control of operations (p = 0.845).

Table 4 shows that in Model 3 of the logistic regression analysis, the answer pattern “incorrect to correct” in knowledge of working environment management was significantly associated with presence of anxiety over radiation exposure after the training session compared to the referent (OR = 4.489; 95% CI = 1.216, 16.571). For Models 1 and 2, no significant factors were associated with anxiety after the training session.

## 4. Discussion

The present study examined the association between anxiety over radiation exposure and possession/acquisition of OHM knowledge regarding working environment management, control of operations, and health management. Contrary to our hypothesis, no significant associations were found between accurate OHM knowledge and absence of anxiety after the training session; instead, acquisition of accurate knowledge of working environment management was significantly associated with presence of anxiety over radiation exposure whereas inaccurate knowledge was not associated with such anxiety. Importantly, acquisition of accurate knowledge of working environment management may lead to mental health problems among OLs, while such accurate knowledge may be a possible contributing factor to the reduction of physical health problems related to radioactivity decontamination work. To promote safe and secure work among radioactivity decontamination workers, both mental health support for OLs who have accurate knowledge of working environment management and educational support for those without are required.

Regarding participant characteristics, age 50-59 years was the most common category, which is consistent with the results of past studies of OLs [12] and radioactivity decontamination workers [13,14,16,17,18]. For the acquisition of the knowledge, the answer pattern “incorrect to correct” was the second most common in working environment management and control of operations, suggesting the effectiveness of the training session. On the other hand, the training session may not have been effective for acquisition of health management knowledge since the second most common pattern was “incorrect to incorrect”.

Answer patterns in knowledge of working environment management were significantly associated with anxiety over radiation exposure after the training session, whereas no significant associations were found before the training session. These results suggest that anxiety over radiation exposure is associated not with possession or lack of accurate knowledge regarding working environment management, but with the acquisition of such knowledge.

Our logistic regression model indicated that the answer pattern “incorrect to correct” in knowledge of working environment management was significantly associated with presence of anxiety over radiation exposure after the training session. Since our study design was cross-sectional, it is impossible to determine the cause-and-effect relationship between answer patterns and presence of anxiety; however, the results of our analysis indicate that the answer patterns for any three knowledge questions were not significantly associated with anxiety before the session (Table 2), whereas the answer patterns in knowledge of working environment management were significantly associated with anxiety after the session (Table 3). Considering these results, acquisition of OHM knowledge may possibly influence the anxiety.

It is of importance that a significant association was found between anxiety over radiation exposure after the training session and accurate knowledge acquisition, specifically in working environment management. Working environment management can be regarded as the most basic measure for prevention of workers’ health impairment in OHM [19]. Working environment management in radioactivity decontamination work consists of tasks related to the group-wide safety of radioactivity decontamination workers; for example, the measurement of ambient dose rate at decontamination sites. Our participants may recognize the risk of radiation exposure for themselves and their colleagues and may realize their responsibilities for radiation prevention. Past studies reported that knowledge of radiation was associated with increased radiation risk perception on health [20], and that positive linear relationships were found between radiation risk perception and anxiety related to radiation [21]. In light of these past studies, it is reasonable to assume that acquisition of accurate knowledge of working environment management of the participants in the current study was associated with the increased radiation risk perception, and such increased radiation risk perception was associated with presence of anxiety over radiation exposure.

Anxiety over radiation exposure backed by accurate knowledge possibly has a double meaning. Anxiety over radiation exposure is perceived as psychologically severe distress [22], and thus may have a negative impact on mental health; however, such anxiety may encourage the avoidance of radiation exposure. Thus, both mental health support for OLs with accurate knowledge of working environment management to reduce the anxiety over radiation exposure, as well as educational support for those with inaccurate knowledge of working environment management to promote work safety and radiation prevention, are required. Our results indicate that anxiety over radiation exposure was prevalent among participants with accurate knowledge of working environment management, whereas no significant associations were found between such anxiety and inaccurate knowledge. In other words, inaccurate knowledge did not significantly increase or decrease anxiety over radiation exposure. Those participants may not be able to fully understand their responsibilities or the risks of their work, and they may therefore not feel the anxiety over radiation exposure. As shown in Table 1, the percentages of participants with correct answers for working environment management, control of operations, and health management after training sessions were 83.9%, 71.3%, and 50.6%, respectively. These results suggest that there may be many OLs without accurate knowledge which should be acquired. We believe that educational support from public administration and occupational health professionals are required.

In Fukushima Prefecture, radioactivity decontamination work in the “difficult-to-return zones” will last until 2023 at the latest [23]. To increase the safety and security of radioactivity decontamination work, it is assumed that OLs as field supervisors are required to make efforts towards reducing the anxiety of their radioactivity decontamination workers by correctly assessing radiation exposure risks based on working environment management, such as measurement of the ambient dose rate of the decontamination sites.

A limitation of the present study was its small sample size. Although the sample size complied with the minimum requirement for statistical analysis, there may have been potential significant associations that could not be revealed in the present study due to the small sample size. In addition, the current study did not include data related to physical aspects of radiation, such as actual dose rate and monitoring situation, and nor did it include data that was possibly associated with anxiety over radiation exposure, such as information from media, colleagues, or family. Future studies should include such data, to obtain more robust evidence regarding the association between anxiety and knowledge. Moreover, the causal relationship between anxiety and knowledge acquisition could not be clarified; subjects’ anxiety may have been aroused by questions before the training session, and they may have searched more deliberately for information in the session due to the aroused anxiety. More sophisticated study designs should help further exploration.

## 5. Conclusions

We revealed that acquisition of accurate knowledge of working environment management was associated with anxiety over radiation exposure. The acquisition of such knowledge may contribute to the reduction of physical health risks, although it may increase the mental health risk. To promote safe and secure work among radioactivity decontamination workers, both mental health support for OLs with accurate OHM knowledge and educational support for those without accurate knowledge are required.

## Figures and Tables

**Table 1 ijerph-17-00228-t001:** Subject Characteristics.

Variables	n (%)
Mean Age ± SD (years)	47.6 ± 12.5
< 30 years	9 (10.3)
30-39 years	13 (14.9)
40-49 years	18 (20.7)
50-59 years	31 (35.6)
≥ 60 years	16 (18.4)
Knowledge of Working Environment Management (Before)	
Correct	60 (69.0)
Incorrect	27 (31.0)
Knowledge of Working Environment Management (After)	
Correct	73 (83.9)
Incorrect	14 (16.1)
Knowledge of Control of Operations (Before)	
Correct	50 (57.5)
Incorrect	37 (42.5)
Knowledge of Control of Operations (After)	
Correct	62 (71.3)
Incorrect	25 (28.7)
Knowledge of Health Management (Before)	
Correct	44 (50.6)
Incorrect	43 (49.4)
Knowledge of Health Management (After)	
Correct	44 (50.6)
Incorrect	43 (49.4)
Anxiety Over Radiation Exposure (Before)	
Very much	1 (1.1)
Somewhat	38 (43.7)
A Little Bit	38 (43.7)
None	10 (11.5)
Anxiety Over Radiation Exposure (After)	
Very Much	3 (3.4)
Somewhat	41 (47.1)
A Little Bit	32 (36.8)
None	11 (12.6)
[Answer Patterns]	
Knowledge of Working Environment Management	
Correct to Correct	56 (64.4)
Correct to Incorrect	4 (4.6)
Incorrect to Correct	17 (19.5)
Incorrect to Incorrect	10 (11.5)
Knowledge of Control of Operations	
Correct to Correct	38 (43.7)
Correct to Incorrect	12 (13.8)
Incorrect to Correct	24 (27.6)
Incorrect to Incorrect	13 (14.9)
Knowledge of Health Management	
Correct to Correct	38 (43.7)
Correct to Incorrect	6 (6.9)
Incorrect to Correct	6 (6.9)
Incorrect to Incorrect	37 (42.5)

**Table 2 ijerph-17-00228-t002:** Associations of anxiety over radiation exposure before training session with age and answer patterns.

Variables	Anxiety Over Radiation Exposure		
	Present	Absent	p-value
Age Group (Years)			0.619
< 30	6 (66.7)	3 (33.3)	
30-39	6 (46.2)	7 (53.8)	
40-49	7 (38.9)	11 (61.1)	
50-59	12 (38.7)	19 (61.3)	
≥ 60	8 (50.0)	8 (50.0)	
Knowledge of Working Environment Management			0.888^a^
Correct to Correct	24 (42.9)	32 (57.1)	
Correct to Incorrect	2 (50.0)	2 (50.0)	
Incorrect to Correct	9 (52.9)	8 (47.1)	
Incorrect to Incorrect	4 (40.0)	6 (60.0)	
Knowledge of Control of Operations			0.75
Correct to Correct	18 (47.4)	20 (52.6)	
Correct to Incorrect	4 (33.3)	8 (66.7)	
Incorrect to Correct	12 (50.0)	12 (50.0)	
Incorrect to Incorrect	5 (38.5)	8 (61.5)	
Knowledge of Health Management			0.642^a^
Correct to Correct	18 (47.4)	20 (52.6)	
Correct to Incorrect	4 (66.7)	2 (33.3)	
Incorrect to Correct	2 (33.3)	4 (66.7)	
Incorrect to Incorrect	15 (40.5)	22 (59.5)	

a—Fisher’s exact test; Note: Associations were examined by the chi-square test.

**Table 3 ijerph-17-00228-t003:** Associations of anxiety over radiation exposure after training session with age and answer patterns.

Variables	Anxiety Over Radiation Exposure		
	Present	Absent	p-value
Age Group (Years)			0.285^a^
< 30	6 (66.7)	3 (33.3)	
30-39	7 (53.8)	6 (46.2)	
40-49	8 (44.4)	10 (55.6)	
50-59	12 (38.7)	19 (61.3)	
≥ 60	11 (68.8)	5 (31.3)	
Knowledge of Working Environment Management			0.040^a^
Correct to Correct	25 (44.6)	31 (55.4)	
Correct to Incorrect	3 (75.0)	1 (25.0)	
Incorrect to Correct	13 (76.5)	4 (23.5)	
Incorrect to Incorrect	3 (30.0)	7 (70.0)	
Knowledge of Control of Operations			0.845
Correct to Correct	21 (55.3)	17 (44.7)	
Correct to Incorrect	5 (41.7)	7 (58.3)	
Incorrect to Correct	12 (50.0)	12 (50.0)	
Incorrect to Incorrect	6 (46.2)	7 (53.8)	
Knowledge of Health Management			0.850^a^
Correct to Correct	19 (50.0)	19 (50.0)	
Correct to Incorrect	3 (50.0)	3 (50.0)	
Incorrect to Correct	2 (33.3)	4 (66.7)	
Incorrect to Incorrect	20 (54.1)	17 (45.9)	

a—Fisher’s exact test; * indicates statistical significance; Note: Associations were examined by the chi-square test.

**Table 4 ijerph-17-00228-t004:** Logistic regression model of associations of anxiety over radiation exposure after training session with age and answer patterns.

Variables	Anxiety Over Radiation Exposure		
	Model 1 OR (95%CI)	Model 2OR (95%CI)	Model 3OR (95%CI)
Age	0.965 (0.686, 1.357)	N/A^b^	0.904 (0.606, 1.351)
Knowledge of Working Environment Management			
Correct to Correct	ref.	ref.	ref.
Correct to Incorrect	3.720 (0.364, 37.99)	3.720 (0.364, 38.038)	10.084 (0.488, 208.436)
Incorrect to Correct	4.030 (1.168, 13.90)^a^	3.968 (1.143, 13.771)	4.489 (1.216, 16.571)*
Incorrect to Incorrect	0.531 (0.124, 2.269)	0.511 (0.116, 2.260)	0.494 (0.104, 2.341)
Knowledge of Control of Operations			
Correct to Correct	ref.	ref.	ref.
Correct to Incorrect	0.578 (0.155, 2.151)	0.560 (0.149, 2.113)	0.478 (0.109, 2.095)
Incorrect to Correct	0.810 (0.291, 2.255)	0.809 (0.290, 2.255)	0.844 (0.281, 2.532)
Incorrect to Incorrect	0.694 (0.196, 2.456)	0.676 (0.189, 2.416)	0.380 (0.087, 1.656)
Knowledge of Health Management			
Correct to Correct	ref.	ref.	ref.
Correct to Incorrect	1.000 (0.179, 5.596)	0.974 (0.170, 5.579)	1.019 (0.155, 6.703)
Incorrect to Correct	0.500 (0.082, 3.063)	0.500 (0.082, 3.066)	0.293 (0.027, 3.130)
Incorrect to Incorrect	1.176 (0.475, 2.914)	1.172 (0.473, 2.905)	0.990 (0.369, 2.659)

a—Statistical significance in this category was not discussed, because no statistical significance was found in the corresponding referent; b—N/A indicates “not applicable”, because age was consistently input in Model 2 and thus the ORs and 95%CI were omitted; * indicates statistical significance by logistic regression analysis.

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
