# Peer review of "Association of Anxiety over Radiation Exposure and Acquisition of Knowledge Regarding Occupational Health Management in Operation Leader Candidates of Radioactivity Decontamination Workers in Fukushima, Japan: A Cross-Sectional Study"

_ijerph, 2019, doi:10.3390/ijerph17010228_

Round 1

Reviewer 1 Report

This is an interesting study supporting earlier evidence that an improvement in knowledge about radiation risks does not necessarily reduce anxiety, but sometimes rather increases it. As the authors themselves write, the number of participants is not very high, but sufficient for this kind of analysis. The increase in anxiety is only observed in the group of respondents going from "inaccurate" to "accurate", i.e. for those whose knowledge is improved, not generally for everybody undergoing the training. That makes sense and is worth reporting. 

I have a few suggestions for changes (which I consider minor, because they don't require further data collection or changes in the analysis):

Lines 19-21:  "In result, acquisition of knowledge of working environment management was significantly associated with presence of anxiety over radiation  exposure after the session compared to possession of the knowledge before and after the session": You probably mean: … acquisition of knowledge… was significantly associated with in increase of anxiety… comparing the situation before and after the session."

Line 37: "Radiation decontamination work": It seems that this expression is used a lot, especially in Japanese publications, but in my view it should be „Radioactivity decontamination work“ or „Radioactive decontamination work"

Lines 53-54: Is this not a repetition of lines 49-52?

Line 101: Maybe I missed it, but didn’t you say the questionnaires were submitted anonymously before and after the session? How did you know „who is who“, i.e. how could you correlated answers before and after?

Lines 114-115: I am not a statistician, so it may just be my ignorance, but I think you should make a few more words explaining the three models. I guess most of your readers would appreciate.

Lines 136-137: Maybe you should mention that there was a trend for young people to show (or admit) more anxiety. I am actually suprised that didn’t come out as significant.

Lines 217-218 "Our results indicate that anxiety over radiation exposure was not prevalent among participants without accurate knowledge.": I don‘t quite understand this statement. You mean anxiety is not more prevalent in these participants than in those with better knowledge?

Author Response

Dear Reviewers,

We wish to express our appreciation to the reviewers for their insightful comments, which have helped us greatly improve our manuscript.

First, we had the manuscript checked by native English-speaking scientific editors, according to the evaluation in “English language and style” from Reviewer 1. Thus, text other than that pointed out by the reviewers has been changed in order to improve the manuscript.

[Reviewer1]

This is an interesting study supporting earlier evidence that an improvement in knowledge about radiation risks does not necessarily reduce anxiety, but sometimes rather increases it. As the authors themselves write, the number of participants is not very high, but sufficient for this kind of analysis. The increase in anxiety is only observed in the group of respondents going from "inaccurate" to "accurate", i.e. for those whose knowledge is improved, not generally for everybody undergoing the training. That makes sense and is worth reporting.

I have a few suggestions for changes (which I consider minor, because they don't require further data collection or changes in the analysis):

Lines 19-21: "In result, acquisition of knowledge of working environment management was significantly associated with presence of anxiety over radiation  exposure after the session compared to possession of the knowledge before and after the session": You probably mean: … acquisition of knowledge… was significantly associated with in increase of anxiety… comparing the situation before and after the session."

Response1: Thank you for your suggestion, according to which we revised the abstract (lines 20-21).

Line 37: "Radiation decontamination work": It seems that this expression is used a lot, especially in Japanese publications, but in my view it should be „Radioactivity decontamination work“ or „Radioactive decontamination work"

Response2: We have now changed “radiation decontamination work” to “radioactivity decontamination work” throughout the manuscript, according to your comment.

Lines 53-54: Is this not a repetition of lines 49-52?

Response3: Reviewer’s comment is correct. We have now removed the text in question.

Line 101: Maybe I missed it, but didn’t you say the questionnaires were submitted anonymously before and after the session? How did you know „who is who“, i.e. how could you correlated answers before and after?

Response4: In the text, we state that “They were handed out two self-administered questionnaires at the beginning of training session” in lines 64-66. We distributed two questionnaires at first, and the participants completed each questionnaire, one before the training sessions and the other after, and then submitted both together. Each pair of questionnaires was held together by a clip, and thus we were able to identify who was whom.

Lines 114-115: I am not a statistician, so it may just be my ignorance, but I think you should make a few more words explaining the three models. I guess most of your readers would appreciate.

Response5: Thank you for your suggestion. To clarify the analysis procedure, we have added the following underlined text to the Statistical Analysis subsection (lines 110–116): Models were built in three steps: a crude model, in which the associations of anxiety over radiation exposure with age and knowledge of OHM were examined without adjustment; an age-adjusted model, in which the association of anxiety over radiation exposure and OHM knowledge was examined, inputting age consistently as an explanatory variable for adjustment; and lastly a multivariate-adjusted model which served as the final model in which all explanatory variables were input simultaneously and fully adjusted.

Lines 136-137: Maybe you should mention that there was a trend for young people to show (or admit) more anxiety. I am actually suprised that didn’t come out as significant.

Response6: Thank you for your suggestion. We have added the following underlined text to the Results section (lines 138-139): ...before the session and variables, although there was trend that the anxiety over radiation exposure was more prevalent in young individuals such as <30 than other age groups.

Lines 217-218 "Our results indicate that anxiety over radiation exposure was not prevalent among participants without accurate knowledge.": I don‘t quite understand this statement. You mean anxiety is not more prevalent in these participants than in those with better knowledge?

Response7: To clarify the meaning, we have added the following underlined text to the Discussion section (lines: 222-225): Our results indicate that anxiety over radiation exposure was prevalent among participants with accurate knowledge of working environment management, whereas no significant associations were found between such anxiety and inaccurate knowledge. In other words, inaccurate knowledge did not significantly increase or decrease the anxiety over radiation exposure.

Reviewer 2 Report

This document has analyzed questionnaire data obtained from the previous study and concluded acquisition of accurate knowledge of working environment management was associated with anxiety over radiation exposure. 

General comments

It is not clear what is difference with previous published paper (Ref 12). Probably, the change in correct knowledge before and after training session has been added. In addition to the questionnaires survey that conducted at the beginning of training session, the questionnaires at the end of training session also have been focused. If so, the document should be modified in order to clarify these points in comparison with the previous published paper (Ref 12). Accurate knowledge of working environment management is a key to reduce anxiety over radiation exposure. However, it is not clear what is knowledge of working environment management. It would be better to attach specific questionnaires in annex. It seems that there is no association between radiation and working environment management. If not, the authors should address what are knowledges of working environment management associated radiation monitoring and dose.

Specific comments

Line 185-187

   The authors discuss anxiety over radiation exposure may be associated not with possession or lack of accurate knowledge, but with acquisition of such knowledge. This discussion would be interesting. If you can find some reference suggesting it, it would be better to strengthen your suggestion. The reference might be available in social psychology.

Line 217-220

    The authors discuss he participants without knowledges may not feel the anxiety over radiation exposure. These knowledges are limited like working environment management, etc.  Information form media would be acquired in daily basis. Some may be involved with having family or not. It would be not easy to conclude from the present  result. 

Author Response

Dear Reviewers,

We wish to express our appreciation to the reviewers for their insightful comments, which have helped us greatly improve our manuscript.

First, we had the manuscript checked by native English-speaking scientific editors, according to the evaluation in “English language and style” from Reviewer 1. Thus, text other than that pointed out by the reviewers has been changed in order to improve the manuscript.

[Reviewer2]

This document has analyzed questionnaire data obtained from the previous study and concluded acquisition of accurate knowledge of working environment management was associated with anxiety over radiation exposure.

<General comments>

It is not clear what is difference with previous published paper (Ref 12). Probably, the change in correct knowledge before and after training session has been added. In addition to the questionnaires survey that conducted at the beginning of training session, the questionnaires at the end of training session also have been focused. If so, the document should be modified in order to clarify these points in comparison with the previous published paper (Ref 12). Accurate knowledge of working environment management is a key to reduce anxiety over radiation exposure. However, it is not clear what is knowledge of working environment management. It would be better to attach specific questionnaires in annex. It seems that there is no association between radiation and working environment management. If not, the authors should address what are knowledges of working environment management associated radiation monitoring and dose.

Response1: We agree with the reviewer’s comment, and believe that it is important to clarify the difference between the current study and our previous study (Reference no 12). The previous study examined what factor affects the possession of accurate knowledge regarding occupational health management; in other words, knowledge was the dependent/objective variable. In contrast, in the current study in which we focused on the workers’ mental health, knowledge was an independent/explanatory variable. Moreover, the current study used data from questionnaires that had been completed both before and after the training session in order to examine the relationship between change of anxiety and knowledge possession, while the previous study used data from after the training session only.  The current study has a different purpose and separate research design from those of our previous study. Thus, we believe that there is no need for comparison between the current and previous studies and that our manuscript should be remain in its current form.

Regarding the reviewer’s question “what is knowledge of working environment management”, we attached the questionnaire actually used in the current study (see attached file for reviewers). This questionnaire is the same as the one used in our previous study, and is included in the Supplementary Materials of our current manuscript. Q1 in the questionnaire was used for assessment of working environment management knowledge. This knowledge is required for operation leaders and is classified as the usage of machines for decontamination-related work, procedure of dose rate measurement at decontamination sites, and related criteria[A]. Q1 consisted of items to assess fundamental knowledge regarding working environment management, according to above-mentioned classification. These items were used and disclosed in our previous study; therefore, we believe that working environment management has been adequately explained in the current manuscript.

Regarding the reviewer’s question “what are knowledges of working environment management associated radiation monitoring and dose”, we could not collect the data on actual dose rate or monitoring situation. Thus, the current study could not examine the association between anxiety over radiation exposure and such data. We have added the following text in the limitations paragraph (lines 241-246): In addition, the current study did not include data related to physical aspects of radiation, such as actual dose rate and monitoring situation, and nor did it include data that was possibly associated with anxiety over radiation exposure, such as information from media, colleagues, or family. Future studies should include such data, in order to obtain more robust evidence regarding the association between anxiety and knowledge.

Reference for reviewer:

[A]Ministry of Health, Labour and Welfare of Japan. Ordinance on Prevention of Ionizing Radiation Hazards at Works to Decontaminate Soil and Wastes Contaminated by Radioactive Materials Resulting from the Great East Japan Earthquake and Related Works. Available online:  http://www.mhlw.go.jp/english/topics/2011eq/workers/ri/rl/rl_130412.pdf

<Specific comments>

Line 185-187

The authors discuss anxiety over radiation exposure may be associated not with possession or lack of accurate knowledge, but with acquisition of such knowledge. This discussion would be interesting. If you can find some reference suggesting it, it would be better to strengthen your suggestion. The reference might be available in social psychology.

Response2: Thank you for your suggestion, and we agree with the reviewer; however, the in-depth discussion for the association between accurate knowledge and increased anxiety has already been included (lines 208–211): “past studies have reported that knowledge of radiation was associated with increased radiation risk perception on health[20], and that positive linear relationships were found between radiation risk perception and anxiety related to radiation[21]”. Any additional discussion to these sentences may be redundant, and thus we respectfully request that this part of our manuscript remain in its current form.

Line 217-220

The authors discuss he participants without knowledges may not feel the anxiety over radiation exposure. These knowledges are limited like working environment management, etc. Information form media would be acquired in daily basis. Some may be involved with having family or not. It would be not easy to conclude from the present 

Response3: We agree with the reviewer’s comment. We have added the following text to limitation paragraph (lines 241–246): In addition, the current study did not include data related to physical aspects of radiation, such as actual dose rate and monitoring situation, and nor did it include data that was possibly associated with anxiety over radiation exposure, such as information from media, colleagues, or family. Future studies should include such data, in order to obtain more robust evidence regarding the association between anxiety and knowledge.